# Effects of Sodium Selenite on the Growth and Photosystem II Activity of *Arthrospira platensis* Gom.

Donghui Gong [1,2,*], Wenxue Wei [1], Ziqing Guo [1], Xiang Ji [3], Xiaoli Zhang [1], Yaxu Yang [1], Shuyu Yu [3], Qingfeng Miao [3], Fucheng Guo [1] and Zhizhong Wang [4,*]

[1] School of Life Science and Technology, UST Inner Mongolia, Baotou 014010, China; weiwx28@163.com (W.W.); guozq1997@139.com (Z.G.); zxl_6018@imust.edu.cn (X.Z.); yyx000404@163.com (Y.Y.); guofucheng1101@163.com (F.G.)

[2] Inner Mongolia Key Laboratory for Biomass-Energy Conversion, Baotou 014010, China

[3] Water Conservancy and Civil Engineering College, Inner Mongolia Agricultural University, Huhot 010019, China; jixiang@imust.cn (X.J.); yushuyu@imau.edu.cn (S.Y.); imaumqf@imau.edu.cn (Q.M.)

[4] Department of Biotechnology, Ordos Vocational College of Eco-Environment, Ordos 017010, China

[*] Correspondence: gongdh1976@163.com (D.G.); 15947515868@126.com (Z.W.)

**Abstract:** *Arthrospira platensis* (*A. platensis*) is a species of cyanobacteria with high economic value; the species is commercially well known as *Spirulina platensis*, and *A. platensis* was used in this paper. Its high adaptability, high photosynthetic efficiency, and fast growth rate make it one of the few cyanobacteria that can be cultivated on a large scale. Therefore, using the selenium enrichment property of *A. platensis* to cultivate selenium-enriched *A. platensis* will not only enhance the physiological efficacy of *A. platensis* but also increase its economic value significantly. In this study, we investigated the effects of sodium selenite on the growth and photosynthetic performance of *A. platensis* selenium by setting different amounts and methods of sodium selenite addition, and we explored the optimal culture conditions of the best dosage and method of sodium selenite addition. The results showed that the experimental group treated with sodium selenite at 700 μmol/L had the fastest growth, and the contents of soluble protein, phycocyanin C, and chlorophyll a increased by approximately 67.9%, 1.44 times, and 38.8% compared to the control group, respectively. Superoxide dismutase (SOD) and catalase (CAT) activity increased by 1.88-fold and 65%, respectively, and malondialdehyde (MDA) levels were reduced by 62% compared to the control group. The results of the OJIP assay showed that the J and I points were significantly higher at the batch addition and treatment concentration of 700 μmol/L, with the rate of QA being reduced and the proportion of the slowly reduced PQ pool being increased. The values of the maximum light energy conversion efficiency (Fv/Fm) per unit of reaction center were higher in both sodium selenite treatment groups than in the control group, indicating that the light energy conversion efficiency of *A. platensis* was promoted under all concentration treatment conditions. The batch addition of sodium selenite at concentrations less than 700 μmol/L resulted in significantly higher ABS/RC values than the control, and they were far superior to the one-time addition method. The reason for this may have been that the batch addition of sodium selenite at low concentrations increased the light absorption capacity of the unit reaction center of PSII, resulting in a rise in captured light energy, a rise in the energy captured by the reaction center for electron transfer (ETo/RC), a decrease in the energy dissipated in the absorption of light energy by the reaction center (DIo/RC), and an increase in the photosynthetic performance index (PI abs).

**Keywords:** *Arthrospira platensis*; *Spirulina platensis*; sodium selenite; soluble protein; chlorophyll a; chlorophyll a fluorescence

## 1. Introduction

*Arthrospira platensis* Gom. (synonym *Spirulina platensis*) is an ancient prokaryotic multicellular filamentous cyanobacteria with a wide distribution range and very high

economic value, and it has been widely used in many fields such as health, food, and medicine. Selenium (Se) is one of the 14 trace elements essential for animal growth and development, and its properties are intermediate between those of metals and nonmetals, with important physiological functions [1]. Selenium exists in human and animal organisms in the form of selenocysteine, which enhances the activity of glutathione peroxidase and catalase, improves the ability of organisms to scavenge reactive oxygen species, alleviates the chain reaction of lipid peroxidation due to effective free radical oxidation, and protects the structure and function of cell membranes [2]. Both a deficiency and an excess of selenium may lead to different diseases, such as decreasing the immunity of the body or causing dense changes in cardiac muscle cells and the abnormal accumulation of lipids and calcium in living organisms. *A. platensis* has an enrichment effect on metal elements which can enrich inorganic selenium and convert it to organic selenium through conversion, thus improving the efficiency of selenium utilization and physiological activity [3]. Selenium-rich *A. platensis* has significantly improved its physiological efficacy compared to traditional *A. platensis*. Li et al. found that selenium-rich *A. platensis* were effective at improving blood phase indices in mice [4]. Dong et al. demonstrated that the consumption of selenium-rich *A. platensis* improved the physiological condition of diabetic mice [5]. Gao et al. found that selenium-rich *A. platensis* peptides have good antihypertensive effects [6], and studies have also reported positive effects of selenium-rich *A. platensis* on anti-fatigue, the enhancement of body immunity, and tumor suppression [7,8]. Selenium in the external environment is transformed into organic selenium in cyanobacteria through the enrichment and transformation of *A. platensis*, and how to coordinate the relationship between selenium enrichment and cyanobacteria growth was the key issue in this study.

Chlorophyll fluorescence kinetic parameters can reflect the energy conversion trajectory of plant photosynthesis [9], which is very sensitive to adversity stress. We can understand the effects of different adversity conditions on plant photosystem II (PSII) activity and the adaptation mechanism of plants to adversity by analyzing the changes in plant chlorophyll a fluorescence parameters under different stress conditions (e.g., drought, high temperature, low temperature, or salt stress) [10]. Some studies have reported the effects of adversity stress on plant growth and PSII activity. Zhang et al. found that salinity stress severely reduced photosystem II performance and inhibited photosynthesis and growth development in rice [11]. Yin et al. demonstrated that high temperature stress led to a reduction in the photosynthetic capacity of PSII in tomato seedlings, and it also limited photosynthetic electron transport in plants [12]. Sun et al. found that treating grapevine roots with low temperatures caused severe damage to leaves and inhibited PSII reaction center activity [13]. Gao et al. conducted an in-depth study on the effect of salt stress on PSII activity in microalgae and found that highly concentrated salt stress conditions resulted in inhibited PSII activity and reduced rates of oxygen release, and the electron transfer rate of the algal cells decreased significantly with the increase in salt concentrations [14]. Various adversity stresses can reduce the photosynthetic activity of plants by inhibiting photosynthetic electron transport [15], which changes the photosynthetic electron transfer rate during PSII activity [16]. In summary, a plant's PSII reaction center is one of the main targets for the action of adversity stress, and an in-depth study on the changes in plant PSII activity under adversity stress is important for analyzing the mechanisms of plant stress resistance and adaptation to adversity. At present, few studies have reported on the effects of sodium selenite on PSII activity in *A. platensis*.

Chen et al. studied the effects of selenium on the antioxidant enzyme activity and photosynthetic pigment content of *A. platensis*. The results showed that the content of carotene was highest when the selenium treatment concentration was 40 mg/L, and the content of photosynthetic pigment significantly decreased compared to the control group when the treatment concentration was higher than 175 mg/L; the antioxidant enzyme activity (SOD, CAT) reached its maximum value when the treatment concentration was 250 mg/L [17]. Zhu et al. found that selenium treatment at a concentration of 1–10 mg/L promoted the growth of *A. platensis*, while treatment at a concentration higher than 50 mg/L

had a significant inhibitory effect on the growth of *A. platensis* [18]. Other scholars' studies have also found that selenium exhibits a phenomenon of "low promotion and high inhibition" on the growth, organic matter accumulation, and pigment content of *A. platensis* [19,20]. Some scholars found that the thylakoids of A. platensis were compacted and the cell wall was disintegrated under high selenium concentration treatment [21]. This study was conducted to investigate the effects of the addition of sodium selenite on the biomass, chlorophyll a content, protein content, antioxidant activity, and photosynthetic activity of PSII in *A. platensis* in order to obtain the best sodium selenite application method and the application amount that could best promote the growth of *A. platensis* so as to obtain high-quality selenium-rich *A. platensis*.

## 2. Materials and Methods

### 2.1. Experimental Materials

The cyanobacteria (*A. platensis*) was from Professor Qiao of the algae research group at Inner Mongolia Agricultural University, and *A. platensis* originated from the alkaline lakes ($39°04'$ N, $109°32'$ E) of the Ordos Plateau in Inner Mongolia [22] (Figure 1).

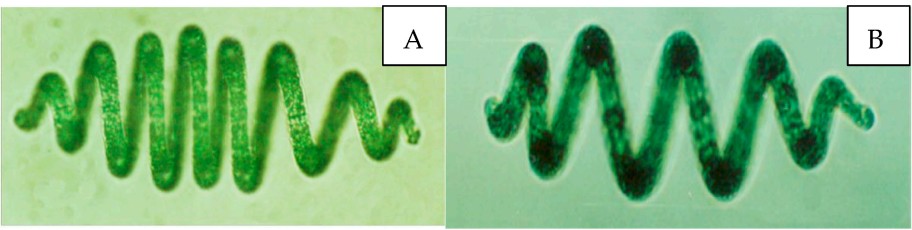

**Figure 1.** Microphotograph picture of *A. platensis*. (**A**,**B**) Filaments are multicellular single-row filaments without heterotypic cells, deep (dark) blue-green, with bubbles. Most of them are spindle-shaped, the number of spirals of algal filaments is 3–12, and most of them are 7–10; the helix width is 32.5~45.0 µm, and the pitch is 32.5~55.0 µm.

### 2.2. Experimental Method

#### 2.2.1. *A. platensis* Culture

Zarrouk standard liquid medium [23] was used and incubated at room temperature under artificial light with intermittent ventilation for a logarithmic growth period with a 12 h:12 h light–dark cycle.

#### 2.2.2. Sodium Selenite Treatment Concentrations

The experimental $Na_2SeO_3$ treatment gradient concentrations are shown in Table 1.

**Table 1.** Concentrations of $Na_2SeO_3$.

| Experimental Group | CK | 1 | 2 | 3 | 4 | 5 |
|---|---|---|---|---|---|---|
| $Na_2SeO_3$ concentration (µmol/L) | 0 | 120 | 300 | 460 | 700 | 880 |

Notes: three parallel samples were set for each experimental group. The $Na_2SeO_3$ treatment concentration was determined by pre-experiments.

#### 2.2.3. Sodium Selenite Addition Method

#### One-Time Addition

The *A. platensis* measurements during the logarithmic growth period were taken, and the initial absorbance of the *A. platensis* solution was adjusted to $OD_{560}$ = 0.25. An $Na_2SeO_3$ mother liquor was added to the culture mediums once at the concentration of each selenium treatment, and the solutions were incubated in 250 mL conical flasks. The culture conditions were as follows: artificial light (fluorescent lamp, 4000 lux) at room temperature, the bottles were shaken 6–8 times per day, a 12 h:12 h light and dark cycle, and three parallel sets for each treatment group.

Additions in Batches

The *A. platensis* was cultured under the same conditions described in Section 2.2.3, and an equal volume of $Na_2SeO_3$ mother liquor was added to the *A. platensis* solutions on days 1 (the initial stage), 5 (the adaptation stage), and 7 (the rapid growth stage), respectively, and the final selenium concentrations were identical to the one-time addition.

### 2.2.4. *A. platensis* Biomass Determination

We prepared gradient concentrations of the *A. platensis* solutions (the $OD_{560}$ values were 0.1, 0.2, 0.3, 0.4, 0.5, 0.6, 0.7, 0.8, and 0.9), and from each concentration of *A. platensis* solution, we took 50 mL, felted it through a glass fiber membrane, and then placed it in an oven at 100 °C overnight. The mass of the filter paper was recorded as L, the mass of the filter paper and the sample after drying the sample dry weight was recorded as A, and the sample dry weight (DW) was the difference between the values of A and L. The relationship between the *A. platensis* liquid concentration ($OD_{560}$) and the sample dry weight (DW) was analyzed by SPASS, and the following regression equation was obtained: $W(g/L) = 0.1049 + 0.503 \times OD_{560}$, and $R^2 = 0.9916$. The *A. platensis* biomasses were calculated using this formula.

### 2.2.5. Determination of the Soluble Protein and Phycocyanin C (C-PC) Contents of the *A. platensis*

The standard curve plotting was completed as follows: We prepared 100 µg/mL of bovine serum protein solution in six glass test tubes. Samples were taken according to Table 2 and mixed thoroughly, and 5 mL of Coomassie brilliant blue (G-250) solution was added to the test tubes, respectively. After shaking well for 5 min, the absorbance values were measured by a spectrophotometer (T6—New Century, Beijing Puxi General Instrument Co., Ltd., Beijing, China) at a wavelength of 595. The protein content and the absorbance were plotted according to the linear regression equations below [24]:

$$y = 0.0009x + 0.0035, R^2 = 0.999 \tag{1}$$

$$PC\ (mg/mL) = 0.187\ OD_{620} - 0.089\ OD_{652}, \tag{2}$$

where PC is the concentration of *A. platensis* C-PC measured in mg/mL.

**Table 2.** The drafting of the standard curves using the Coomassie brilliant blue method.

| Tube Number | 1 | 2 | 3 | 4 | 5 | 6 |
|---|---|---|---|---|---|---|
| Standard protein solution (mL) | 0 | 0.2 | 0.4 | 0.6 | 0.8 | 1 |
| Distilled water content (mL) | 1 | 0.8 | 0.6 | 0.4 | 0.2 | 0 |
| Protein content (ug) | 0 | 20 | 40 | 60 | 80 | 100 |

The determination of the samples was completed as follows: We took 10 mL of sample and centrifuged (TG16-WS, Hunan Xiangyi Laboratory Instrument development Co., Ltd., Changsha, China) it at 12,000 r/min for 20 min. The supernatant was discarded, the *A. platensis* sludge was collected, and phosphate-buffered solution (PBS, 0.1 mol/L, pH = 7.0) was added to wash the sample 3 times. The obtained *A. platensis* cells were evenly mixed with PBS (0.05 mol/L, pH = 7.0), freeze–thaw cycles were repeated 4 times, and then the extracts were centrifuged (8000 r/min for 10 min, TG16-WS, Hunan Xiangyi Laboratory Instrument development Co., Ltd., Changsha, China) and 5 mL of G-250 solution was added to the supernatant. The solution was mixed thoroughly and stored for 2 min. The absorbance values at wavelengths of 595 nm, 620 nm, and 652 nm were measured, and the soluble protein content was calculated according to Equation (1) and the C-PC content was calculated according to Equation (2).

2.2.6. Determination of Chlorophyll A Content in the *A. platensis*

We took 10 mL of sample and centrifuged it (12000 r/min for 20 min, TG16-WS, Hunan Xiangyi Laboratory Instrument development Co., Ltd., Changsha, China). The supernatant was discarded, the *A. platensis* sludge was collected, and 10 mL of 96% ethanol was added. The solution was placed in a refrigerator at 4 °C under dark conditions for 30 h. The extract was then centrifuged (10,000 r/min for 8 min, TG16-WS, Hunan Xiangyi Laboratory Instrument development Co., Ltd., Changsha, China) and the supernatant was measured with an microplate reader (Multiskan FC, Thermo Labsystems Co., Ltd., Middleton, WI, USA) at 663 nm, with OD values of 645 nm [25]. The concentration of chlorophyll a was then calculated according to Equation (3):

$$\text{Chl.a} = 12.7 \times \text{OD}_{663} - 2.96 \times \text{OD}_{645} \tag{3}$$

2.2.7. Determination of the Antioxidant Enzyme Activity in the *A. platensis*
Determination of Superoxide Dismutase (SOD)

The superoxide dismutase (SOD) activity of the *A. platensis* was measured using the nitrogen blue tetrazolium (NBT) method [26]. A total of 3 mL of reaction solution was put in a test tube with good transparency and mixed well. The reaction solution composition consisted of 0.05 mol/L of PBS (1.5 mL (pH = 7.8)), 130 mmol/L of methionine (Met) solution (0.3 mL), 750 µmol/L of nitrogen blue tetrazolium (NBT) solution (0.3 mL), 100 µmol/L of disodium EDTA (0.3 mL), 20 µmol/L of riboflavin (0.3 mL), 0.05 mL of enzyme extraction solution, and 0.25 mL of distilled water. Two control tubes were set up at the same time, and in both control tubes, PBS was used instead of the enzyme extraction solution, and one of the control tubes was placed in the dark while the other control tube and each of the remaining tubes were placed under 4000 lx daylight for 20 min. After the reaction was completed, a blank control tube placed in a dark place was used as a blank for zeroing, and the $\text{OD}_{550}$ value of each sample was measured. The SOD activity (U/mg protein) was calculated according to Equation (4):

$$\text{SOD activity} = ((OD_{ck} - OD_s) \times V_T)/0.5 \times OD_{ck} \times W \times V_S \times P, \tag{4}$$

where $OD_{ck}$ is the absorbance value of the light control, $OD_s$ is the absorbance value of the sample tube, $V_T$ is the total volume of the extraction solution (mL), $V_S$ is the volume of the extraction solution at the time of determination (mL), $W$ is the biomass of the extracted enzyme solution of the *A. platensis* (g/L), and $P$ is the protein content of the *A. platensis* cells (mg/L).

Measurement of Catalase (CAT)

The catalase (CAT) activity was determined using the UV spectrophotometric method [26]. We placed 0.1 mL of enzyme extract and 1.9 mL of 0.1 mol/L PBS (pH = 7.0) in a test tube, and 1 mL of 0.1 mol/L $H_2O_2$ was added to the tube. Then, counting was started, absorbance was rapidly measured at 240 nm, and readings were taken at 1 min intervals for 4 min while the control tubes were measured with PBS instead of enzyme extract. The decrease in $\text{OD}_{240}$ by 0.1 per minute was used as 1 unit of enzyme activity (U/mg protein). The CAT activity (U/mg protein) was calculated according to Equation (5):

$$\text{CAT activity} = ((OD_{ck} - OD_s) \times V_T)/0.1 \times t \times W \times V_S \times P, \tag{5}$$

where $OD_{ck}$ is the absorbance value of the control, $OD_s$ is the absorbance value of the sample tube, $V_T$ is the total volume of the extraction solution (mL), $V_S$ is the volume of the extraction solution at the time of determination (mL), $W$ is the biomass of the extracted enzyme solution of the *A. platensis* (g/L), and $P$ is the protein content of the *A. platensis* cells (mg/L).

Determination of Malondialdehyde (MDA)

The thiobarbituric acid method was used to determine the MDA content [26]. We placed 1 mL of enzyme extract in a test tube and 3 mL of 0.67% thiobarbituric acid (TBA) solution was added. The absorbance of the supernatant in each tube was measured at absorbance values of 450 nm, 532 nm, and 600 nm, respectively, and 0.67% TBA solution was used as a blank. The MDA content (μmol/mg of protein) was calculated according to Equations (6) and (7):

$$C_{MDA} = 6.45(OD_{532} - OD_{600}) - 0.56OD_{450} \tag{6}$$

$$MDA\ content = (C_{MDA} \times V_T)/(W \times 1000 \times P), \tag{7}$$

where $C_{MDA}$ is the MDA concentration of the enzyme extract (μmol/L); $V_T$, $W$, and $P$ have the same meanings as in Equation (5).

### 2.2.8. Measurement of the Chlorophyll A Fluorescence Parameters

The concentration of the *A. platensis* was adjusted to $OD_{560} = 0.05$ and the fluorescence parameters of the chlorophyll a in the samples were determined using a plant efficiency meter (Handy PEA, Hansatech, UK) after 20 min of dark adaptation.

### *2.3. Data Processing*

The experimental data were processed using SPSS 26.0 (IBM SPSS Statistics 26.0).

## 3. Results and Discussion
### *3.1. Effect of Different $Na_2SeO_3$ Concentration Treatments on the Growth of A. platensis*

As shown in Figure 2, the growth of *A. platensis* was promoted by the addition of different concentrations of $Na_2SeO_3$. Under the different treatment concentrations of the $Na_2SeO_3$, the biomasses showed gradual increases with the increases in treatment time in the different treatment groups. The growth rates of all the treatment groups were higher than that of the control, and the biomasses were highest at additions (the one-time additions) of $Na_2SeO_3$ treatment up to the fourteenth day of incubation at a selenium concentration of 700 μmol/L. This increased the biomass by 25% compared to the control group, and the difference was significant ($p < 0.05$). There was no significant difference in the sodium selenite concentration between the 120 and 460 μmol/L treatment groups ($p > 0.05$). The $Na_2SeO_3$ was added in batches (Figure 2B), and the biomasses of *A. platensis* increased by more than 30% compared to the control group when the concentrations of $Na_2SeO_3$ were 120–460 μmol/L on the fourteenth day of incubation. The concentration of $Na_2SeO_3$ at 880 μmol/L had the least effect on the growth of the cyanobacteria. Comparing the growth curves of the two groups, the biomasses of each experimental group with additions in batches were higher than those of the experimental groups with one-time additions for the same treatment time.

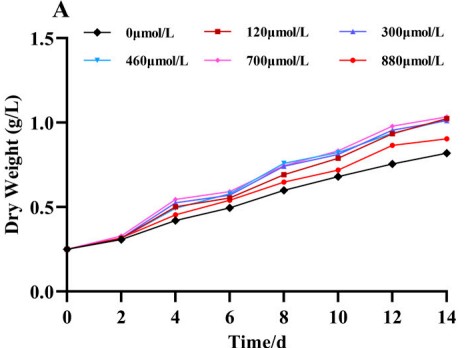
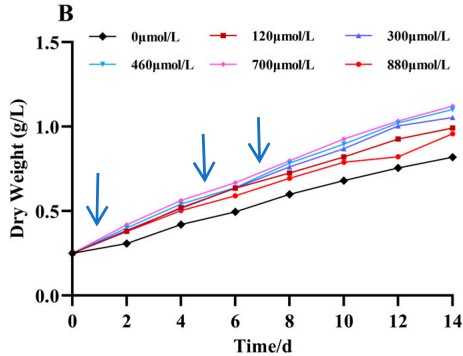

**Figure 2.** Effect of $Na_2SeO_3$ on the growth of *A. platensis*. (**A**) One-time addition; (**B**) batch additions. Arrows indicate the time of sodium selenite supplementation.

*3.2. Effect of Different Na$_2$SeO$_3$ Concentration Treatments on the Soluble Protein Content of the A. platensis*

The changes in the intracellular soluble protein content of the *A. platensis* treated with different concentrations of Na$_2$SeO$_3$ are shown in Figure 3. As shown in the figure, the soluble protein contents changed in a single-peak curve (one-time addition) with the increases in Na$_2$SeO$_3$ concentration on the fourteenth day. The protein content was highest when the exogenous Na$_2$SeO$_3$ treatment concentration was 700 μmol/L, and the soluble protein in the cells of the samples increased by 17% compared with the control, but the differences between the treated groups and the control group were not significant ($p > 0.05$) for each concentration (Figure 3A). When the Na$_2$SeO$_3$ was added in batches (Figure 3B), the changes in the soluble protein in each concentration treatment group also showed a trend of increasing and then decreasing, with the highest point appearing in the experimental group with a concentration of 700 μmol/L. This group had the highest soluble protein content, with an increase of approximately 67.9% compared with the control group. The concentration of the Na$_2$SeO$_3$ was higher than 700 μmol/L for this group and the soluble protein in the *A. platensis* cells gradually decreased, but it remained significantly higher than the control group ($p < 0.05$).

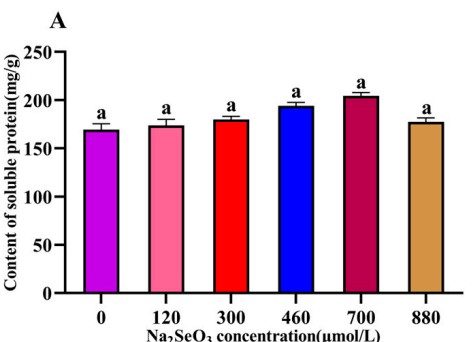 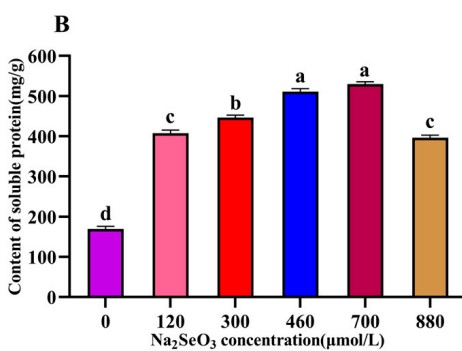

**Figure 3.** Effect of Na$_2$SeO$_3$ on the content of soluble protein of *A. platensis*. (**A**) One-time addition; (**B**) batch additions. Note: data with different lowercase letters in the legend indicate significant differences ($p < 0.05$).

*3.3. Effect of Different Na$_2$SeO$_3$ Concentrations on the C-PC Content of the A. platensis*

The changes in the intracellular C-PC content of the *A. platensis* treated with different concentrations of Na$_2$SeO$_3$ are shown in Figure 4. The changes in the C-PC contents with the increases in Na$_2$SeO$_3$ concentration were the same as those shown in Figure 4 on the fourteenth day of sample treatment. The highest C-PC content was observed when the Na$_2$SeO$_3$ concentration was 700 μmol/L, and the C-PC in the sample cells increased by 45% compared with the control; the differences between the treatment groups and the control group were significant ($p < 0.05$) for each concentration (Figure 4A). When the Na$_2$SeO$_3$ was added in batches (Figure 4B), the changes in the C-PC content in each concentration treatment group also showed an increasing trend, followed by a decreasing trend, and the highest point was found in the experimental group with a concentration of 700 μmol/L; this group had the highest C-PC content, with an increase of approximately 1.44 times compared to the control group. The C-PC contents in the *A. platensis* cells gradually decreased as the concentrations of Na$_2$SeO$_3$ rose to higher than 700 μmol/L.

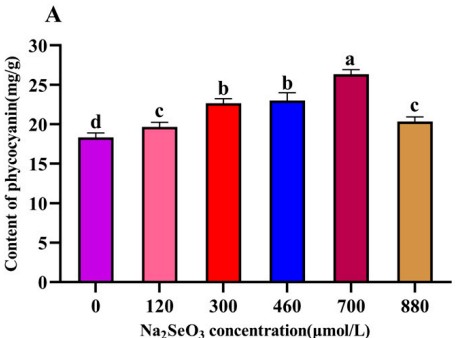
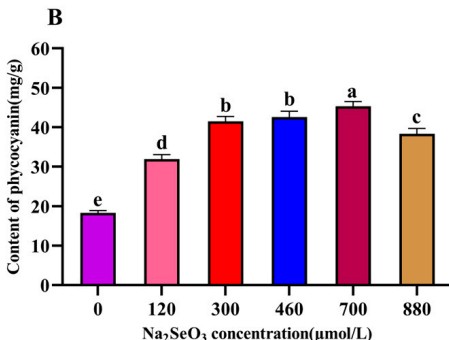

**Figure 4.** Effect of Na₂SeO₃ on the content of C-PC of *A. platensis*. (**A**) One-time addition; (**B**) batch additions. Note: data with different lowercase letters in the legend indicate significant differences ($p < 0.05$).

### 3.4. Effect of Different Na₂SeO₃ Concentration Treatments on the Chlorophyll A Content of the A. platensis

The chlorophyll a content in plants is closely related to their photosynthesis ability and their growth speed. The variation trend for chlorophyll a content in the *A. platensis* showed a single-peak curve, which was the same as the trends for soluble protein and C-PC contents of the *A. platensis*. The peak occurred in the 700 μmol/L (one-time addition) treatment group (Figure 5A), which was near double that of the control group, and the chlorophyll a contents in the 120 μmol/L and 300 μmol/L treatment groups were not significantly different from that of the control group ($p > 0.05$). The chlorophyll a contents of the samples were significantly lower ($p > 0.05$) compared to the peak for the group with the treatment concentration of 880 μmol/L. The most significant increases in chlorophyll a contents were observed for the 460 μmol/L and 700 μmol/L Na₂SeO₃ treatments compared to the control group when the Na₂SeO₃ was added in batches (Figure 5B) ($p > 0.05$), and the chlorophyll a contents of the samples gradually decreased when the concentrations were higher than 700 μmol/L.

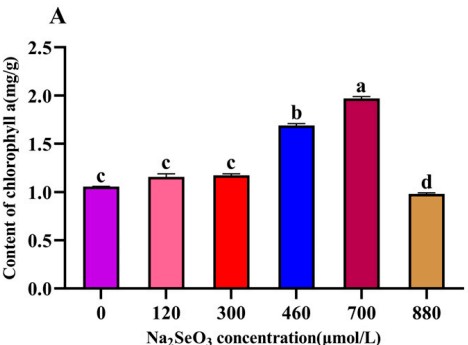
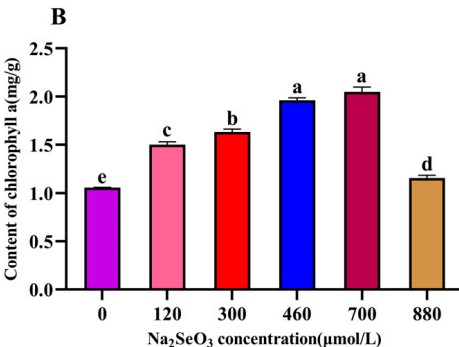

**Figure 5.** Effect of Na₂SeO₃ on the content of chlorophyll a of *A. platensis*. (**A**) One-time addition; (**B**) batch additions. Note: data with different lowercase letters in the legend indicate significant differences ($p < 0.05$).

### 3.5. Effect of Different Na₂SeO₃ Concentration Treatments on the Superoxide Dismutase (SOD) Activity in the A. platensis

The effects of the different concentrations of Na₂SeO₃ treatment on the superoxide dismutase activity of the *A. platensis* are shown in Figure 6. The SOD activity of the *A. platensis* treated with the different concentrations of Na₂SeO₃ showed a trend of increasing and then decreasing with the increase in treatment concentration on the fourteenth day. As shown in Figure 6A, the highest SOD activity was observed in the experimental group when the Na₂SeO₃ treatment concentration was 700 μmol/L (one-time addition), and the activity was nearly double that of the control group. The SOD activity was significantly higher in

all treatment groups than in the control group ($p > 0.05$), and the SOD activity of the sample treated with $Na_2SeO_3$ at a concentration of 880 µmol/L was significantly lower than the peak activity ($p > 0.05$). When the $Na_2SeO_3$ was added in batches (Figure 6B), the SOD activity increased more significantly ($p > 0.05$) at the concentrations of 460 µmol/L and 700 µmol/L compared to the control group, and the SOD activity of the samples gradually decreased when the concentrations were higher than 700 µmol/L, though the activity remained significantly higher ($p > 0.05$) than that of the one-time addition group.

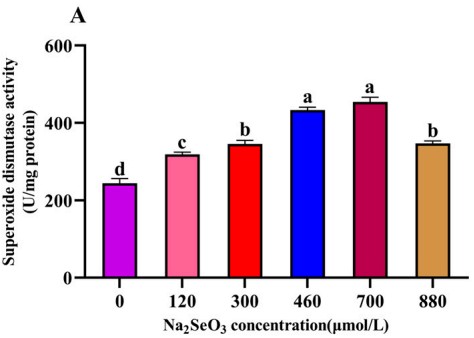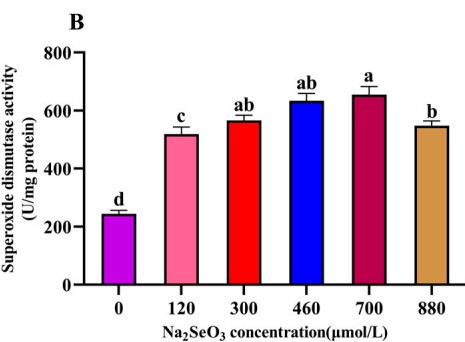

**Figure 6.** Effect of $Na_2SeO_3$ on the superoxide dismutase activity of *A. platensis*. (**A**) One-time addition; (**B**) batch addition. Note: data with different lowercase letters in the legend indicate significant differences ($p < 0.05$).

*3.6. Effect of Different $Na_2SeO_3$ Concentration Treatments on the Catalase (CAT) Activity of the A. platensis*

The variation trends of the different concentrations of $Na_2SeO_3$ treatment on the activity of catalase (CAT) in the *A. platensis* (Figure 7) were consistent with those of the SOD activity, and the highest values were found in the 700 µmol/L treatment group. The CAT activity was significantly higher in the treatment groups (120–700 µmol/L) than in the control group ($p > 0.05$), and the CAT activity of the samples decreased gradually at concentrations higher than 700 µmol/L compared to the peak activity ($p > 0.05$).

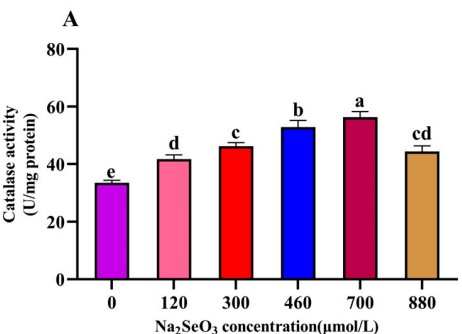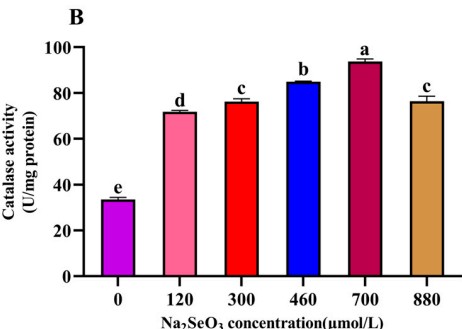

**Figure 7.** Effect of $Na_2SeO_3$ on the catalase activity of *A. platensis*. (**A**) One-time addition; (**B**) batch addition. Note: data with different lowercase letters in the legend indicate significant differences ($p < 0.05$).

*3.7. Effect of Different $Na_2SeO_3$ Concentrations on the Malondialdehyde Content (MDA) in the A. platensis*

The results of the different $Na_2SeO_3$ treatments on the malondialdehyde (MDA) contents of the *A. platensis* are shown in Figure 8. The MDA content of the *A. platensis* treated with different concentrations of $Na_2SeO_3$ on the fourteenth day showed a trend of decreasing and then increasing with the increases in treatment concentration. The experimental group with the lowest MDA content was found at the concentration of 700 µmol/L (for both the one-time addition and batch addition), and the MDA contents decreased by 25% (one-time addition) and 62% (batch addition) compared with the control

group, respectively. The MDA contents were significantly lower in all treatment groups (120–880 μmol/L) compared to the control group ($p > 0.05$).

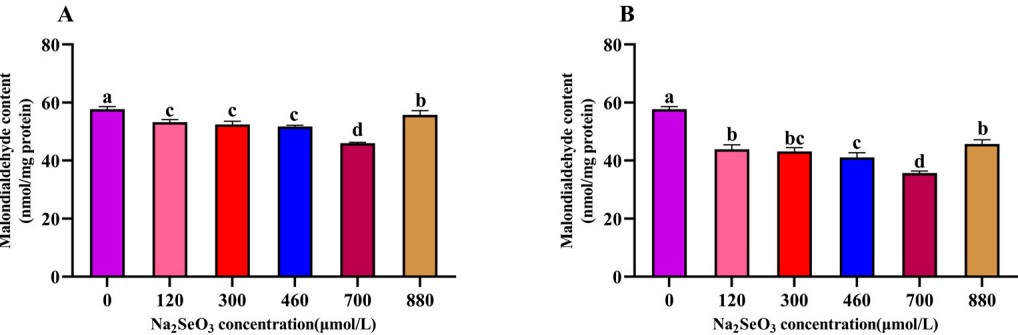

**Figure 8.** Effect of $Na_2SeO_3$ on the malondialdehyde content of *A. platensis*. (**A**) One-time addition; (**B**) batch addition. Note: data with different lowercase letters in the legend indicate significant differences ($p < 0.05$).

*3.8. Effect of Different $Na_2SeO_3$ Concentrations on the Chlorophyll A Fluorescence Induction Kinetic Curves of the A. platensis*

The changes in the chlorophyll a fluorescence of the *A. platensis* cells treated with different concentrations of $Na_2SeO_3$ are shown in Figure 9. The O-J-I-P curves of each treatment group at the different treatment concentrations (one-time addition, Figure 9A) essentially overlapped, and the chlorophyll a fluorescence intensity between the samples was not significantly different from that of the control group ($p > 0.05$). The shapes of the curves changed more significantly under the different concentration gradient treatments when the $Na_2SeO_3$ was added in batches (Figure 9B). The relative fluorescence intensity changes in the groups did not show significant differences at point O, but the fluorescence values of the treated group with treatment concentration of 700 μmol/L were higher than those of the other treated groups at points J and I ($p > 0.05$).

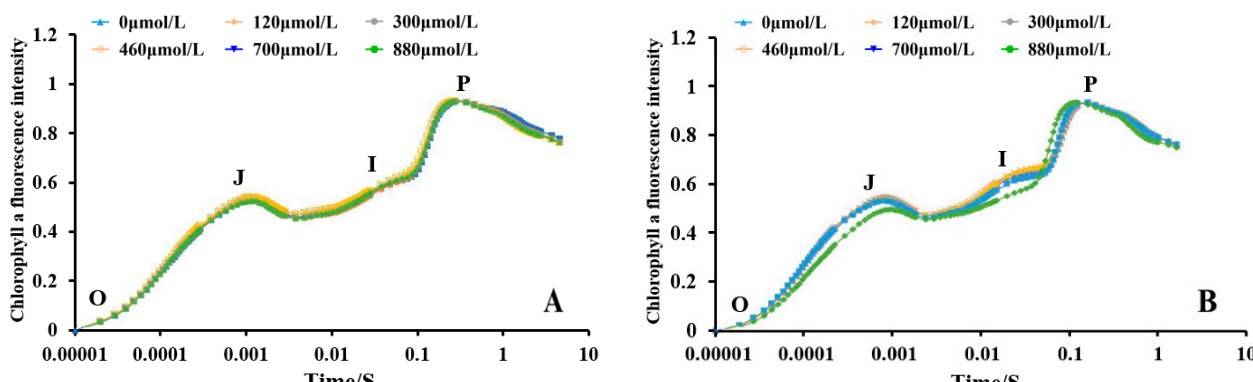

**Figure 9.** Effect of different $Na_2SeO_3$ concentrations on the chlorophyll a fluorescence induction kinetic curves of *A. platensis*. (**A**) One-time addition; (**B**) batch addition.

*3.9. Comparison of the Parameters of the Fluorescence Induction Kinetic Curves of the A. platensis at Different Concentrations of $Na_2SeO_3$*

For the chlorophyll a fluorescence induction kinetic curve parameters, the maximum light energy conversion efficiency per unit reaction was expressed using Fv/Fm. The Fv/Fm values for each treatment group increased significantly compared to the control group after the fourteenth day of incubation at the different $Na_2SeO_3$ treatment concentrations (one-time addition, Figure 10A). The Fv/Fm values reached their highest at the treatment concentration of 700 μmol/L, and the values increased by approximately 9% (Figure 10A) and 10% (Figure 10B) compared to the control group, respectively. The values of the light energy absorbed per unit of reaction center (ABS/RC) were higher in all treatment groups

than they were in the control group, with the highest ABS/RC value in the experimental group treated at a concentration of 700 μmol/L (an increase of approximately 2% compared to the control group). The change in light energy absorption in the reaction center of each $Na_2SeO_3$ concentration treatment group showed a single-peak curve, and the absorbed light energy reached its peak at the treatment concentration of 700 μmol/L when the $Na_2SeO_3$ was added in batches (Figure 10B) (an increase of 20.7% compared to the control group). When the concentrations of $Na_2SeO_3$ were higher than 700 μmol/L, the ABS/RC values gradually decreased.

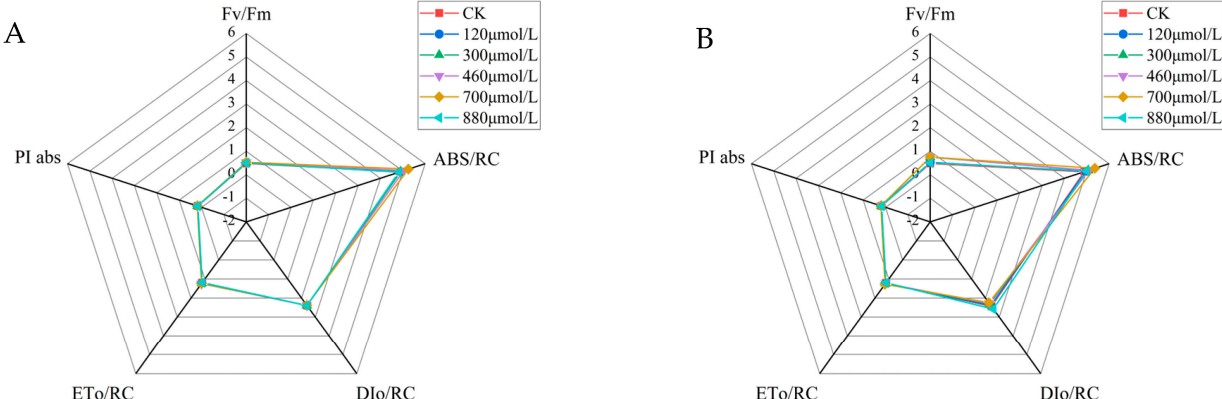

**Figure 10.** Comparison of the parameters of the fluorescence induction kinetics curves of the *A. platensis* at different $Na_2SeO_3$ concentrations. (**A**) One-time addition; (**B**) batch addition.

The one-time additions of the different $Na_2SeO_3$ treatment concentrations had no significant effects on the heat dissipation (DIo/RC) in the photosystem II reaction centers (Figure 10A). The DIo/RC values showed gradual decreases with the increases in $Na_2SeO_3$ concentrations using batch additions (Figure 10B), and they were significantly lower ($p > 0.05$) than those of the control group when the treatment concentrations were 460–700 μmol/L. When the treatment concentration was 880 μmol/L, the DIo/RC value of the reaction center increased by 7.5% compared to the control group, and the difference was significant ($p > 0.05$). The values of energy captured by the *A. platensis* PSII reaction center for QA reductions (ETo/RC) were progressively higher than those of the control group for the one-time additions of the different $Na_2SeO_3$ treatment concentrations, and the experimental group treated with 700 μmol/L had the highest ETo/RC values, with increases of approximately 3% compared to the control group (Figure 10A). The highest ETo/RC values were reached at treatment concentrations of 700 μmol/L when the $Na_2SeO_3$ was added in batches (Figure 10B), with an approximately 6% increase compared to the control. The highest values of the photosynthetic performance index (PI abs) for the PS II reaction centers were achieved by the 700 μmol/L (one-time addition and batch addition) treatment groups, with increases of approximately 4% and 14% compared to the control group, respectively.

## 4. Discussion

### 4.1. Effect of $Na_2SeO_3$ on the Growth, Chlorophyll A, and Protein Contents of the A. platensis

In this study, we investigated the effects of exogenous selenium on the growth and photosynthetic performance of *A. platensis* by setting different concentrations of sodium selenite additions and using different addition methods, and we investigated the optimal culture conditions for selenium-rich *A. platensis*. The results showed that the growth of *A. platensis* was promoted by the different concentrations of $Na_2SeO_3$ and the addition methods. $Na_2SeO_3$ additions at concentrations of 300–700 μmol/L had good promotion effects on the growth of the *A. platensis*. Under the same treatment conditions, the biomass of each treatment group with the batch additions were higher than those of the one-time additions. The reason for this may be that the fractionated addition mitigated the effects

of the high exogenous selenium concentrations on the *A. platensis* metabolism, which improved the adaptability of the cyanobacteria to the exogenous selenium treatment. The chlorophyll a and protein contents of the *A. platensis* increased significantly ($p > 0.05$) compared with the control group when the concentrations of the $Na_2SeO_3$ treatments were 460–700 μmol/L, and the contents decreased when the concentrations were higher than 700 μmol/L, which showed the phenomena of low promotion and high inhibition. The causes of these phenomena were related to the high concentration of exogenous selenium inhibiting the synthesis of *A. platensis* cellular proteins and chlorophyll a or accelerating the degradation process. It was shown that the appropriate concentration of exogenous selenium would promote the growth of *A. platensis* while high concentrations of exogenous selenium stress increased the chlorophyllase activity and promoted the degradation of chlorophyll a, and the decreases in chlorophyll a contents would lead to decreases in the photosynthetic capacity of the *A. platensis*, resulting in the inhibition of growth [27]. A similar phenomenon was found by Gong et al. [28] in their study on the effect of salt stress on the photosynthesis of *A. platensis*. There are certain differences in the tolerance and enrichment ability of different algae to selenium. Ye et al. [29] conducted in-depth research on the effects of similar methods on the organic selenium conversion ability of *Porphyridium cruentum*, and the results showed that within the concentration range of 0.2 and 0.5 mg/L, the biomass of Purple Chlorella was directly proportional to the added selenium concentration. However, the biomass significantly decreases at high concentrations of selenium. Cao et al.'s [30] study results showed that the biomass and pigment content of Chlorella decreased with the increase in sodium selenite concentration and showed different physiological tolerance during the cultivation process. The results of *Caulerpa lentillifera* showed that 2.0 mg/L sodium selenite was suitable for selenium-rich cultivation with the highest biomass and photosynthetic pigment content [31]. It was found from the research reports that different algae had different enrichment and tolerance of sodium selenite, but the overall performance had the effect of low promotion and high inhibition [29–31].

*4.2. Effect of $Na_2SeO_3$ on the Antioxidant Enzyme Activity of A. platensis*

Superoxide dismutase (SOD) and catalase (CAT) are the body's primary substances for scavenging free radicals, and they play crucial roles in the oxidative and antioxidant balances in the body [32]. SOD can catalyze the dismutation of superoxide into oxygen and $H_2O_2$. Catalase (CAT) is a cytoplasmic antioxidant enzyme commonly found in living organisms, and its main function is to catalyze the decomposition of $H_2O_2$ into $O_2$ and $H_2O$. CAT can protect the integrity of the cell membrane system and a cell's structure [33] and delay cellular senescence, as well as tissue browning [34]. CAT and SOD are both important scavengers of reactive oxygen species in living organisms, which are essential for plants to cope with adverse environmental stresses [35]. They are also key regulators of reactive oxygen species homeostasis in plant cells [36]. The concentrations of $Na_2SeO_3$ in the range of 120–700 μmol/L had good promotion effects on the SOD and CAT activity in *A. platensis*, and the enzyme activities of the batch addition treatments were higher than those of the one-time additions. This phenomenon indicated that the low concentrations of $Na_2SeO_3$ could promote the antioxidant capacity of a cell, which was beneficial to the growth of the cyanobacteria cells. However, the enzyme activity gradually decreased at $Na_2SeO_3$ concentrations higher than 700 μmol/L. This indicated that too high concentrations of $Na2SeO_3$ stress led to reduced defense abilities of the antioxidant enzyme systems in the *A. platensis* cells, as well as the excessive accumulation of $H_2O_2$, the oxidation of cell membranes by the ROS, and the formation of lipid peroxidation products, which damaged the *A. platensis* cells due to the effect of heavy metal [37]. Malondialdehyde (MDA) is the final breakdown product of membrane lipid peroxidation, and changes in MDA contents can reflect the extent of tissue damage during lipid peroxidation and scavenging of the reactive oxygen species in the body. Under the same concentration treatment conditions, the MDA contents of each treatment group with the batch additions

were lower than those of the one-time addition treatment groups, and the concentrations of $Na_2SeO_3$ between 120 and 700 μmol/L effectively inhibited increases in the reactive oxygen contents of the *A. platensis*, with the most significant at the concentration of 700 μmol/L. When the concentrations of $Na_2SeO_3$ were higher than 700 μmol/L, the MDA contents of the *A. platensis* cells increased significantly compared to the other treatment groups. This indicated that the appropriate low dosage and addition of $Na_2SeO_3$ could inhibit lipid peroxidation. Excessive concentrations of $Na_2SeO_3$ exhibited certain levels of toxicity, which increased the contents of the reactive oxygen species in the cyanobacteria cells, thus disrupting the dynamic balance of reactive oxygen species production and removal in the cells and exceeding the defense limits of the antioxidant enzymes, as well as accumulating a large amount of reactive oxygen species, which led to a gradual increase in the MDA content. Similar results were obtained from the studies on *Porphyridium* sp., *Dunaliella salina*, and *Chlorella pyrenoidosa*. Appropriate selenium concentration treatment could increase the activity of antioxidant enzymes and decrease the content of MDA in algal cells [38–40].

*4.3. Effect of $Na_2SeO_3$ on the Chlorophyll A Fluorescence of the A. platensis*

Photosystem II (PSII) plays an important role in photosynthesis in the absorption, transfer, and conversion of light energy, and it is closely related to adversity stress. Changes in PSII reaction center parameters under adversity stress reflect a plant's response to adversity, and these are important for analyzing the mechanisms of a plant's regulation and adaptation to adversity conditions [41]. There were no significant differences in the fluorescence values of the treatment groups for the one-time addition treatment ($p > 0.05$). The chlorophyll a fluorescence intensities at different $Na_2SeO_3$ concentrations were higher at the J and I points of the treated groups than in the control group for the batch addition treatments, and the fluorescence values of the treatment group with the concentration of 700 μmol/L were higher than those of the other groups at the J and I points, but the differences between the treatment groups were not significant ($p > 0.05$). The rise in values at the J points indicated increases in the rates at which QA was being reduced, and the rises in the I-point values indicated increases in the proportion of slowly reduced PQ pools, suggesting that excessive $Na_2SeO_3$ concentration stress restricted electron transfer from QA to QB. The maximum light energy conversion efficiency (Fv/Fm) values per unit of reaction center were higher in each treatment group than in the control group, which indicated that the light energy conversion efficiency of the *A. platensis* was promoted under all concentration treatment conditions. These results were consistent with the results for the growth rates and chlorophyll a and protein contents of the *A. platensis*. The absorbed light energy values per reaction center (ABS/RC) for each concentration treatment group using the one-time additions of $Na_2SeO_3$ were significantly higher than those of the control group ($p > 0.05$), and the ABS/RC values were significantly higher than the control when the concentrations were less than 700 μmol/L. The reason for this may have been due to the fact that the addition of $Na_2SeO_3$ in batches at low concentrations increased the light absorption capacity of the PSII unit reaction centers. It may also have been due to the synthesis of pigments being promoted, which led to the captured light energy being increased. Therefore, the energy captured by the reaction center for electron transfer (ETo/RC) also increased, the energy dissipated in the absorption of the light energy by the reaction centers (DIo/RC) decreased, and the photosynthetic performance index (PI abs) improved. Conversely, treatment with high concentrations of exogenous $Na_2SeO_3$ degraded or inactivated the reaction centers and increased the heat dissipation energy (DIo/RC), which reduced the energy used for electron transfer (ETo/RC) and reduced the photosynthetic performance index (PI abs).

There are certain differences in PSII response and adaptation mechanisms among different algae under selenium stress. Studies on the metabolic effects of different doses of sodium selenite showed that no inhibitory effect was shown when the concentration of sodium selenite was 15 mg/L, but the growth and photosynthesis of *Anabaena* were significantly inhibited when the concentration was 20–100 mg/L; the photosynthetic efficiency

of PSII and the electron transport rate of *Anabaena* increased when the concentration was in the range of 10–15 mg/L, which indicated that the photosynthesis was improved [42]. Research has found that the addition of Se nanocarboxylates at smaller concentrations (0.07 or 0.2 mg $L^{-1}$) at first caused the retardation of *Chlorella vulgaris* growth, but that effect disappeared after 18–24 days of cultivation. The addition of 0.4–4 mg $L^{-1}$ of Se nanocarboxylates caused the evident initial increase in such chlorophyll a fluorescence parameters as maximal quantum yield of photosystem II photochemistry (Fv/Fm) and the quantum yield of photosystem II photochemistry in the light-adapted state (Fv'/Fm'). Photochemical fluorescence quenching coefficients increased after 6 days of the addition of 2 or 4 mg $L^{-1}$ of Se nanocarboxylates. Those alterations affected the overall quantum yield of the photosynthetic electron transport in photosystem II [43].

## 5. Conclusions

The addition of exogenous $Na_2SeO_3$ in batches was superior to the one-time addition method in terms of the growth, accumulation of organic materials, and improved antioxidant activity of *A. platensis*. The treatment with suitable low concentrations of $Na_2SeO_3$ improved the maximum light energy conversion efficiency (Fv/Fm), light energy absorption (ABS/RC), energy of electron transfer (ETo/RC), and photosynthetic performance index (PI abs) per unit of reaction center of the *A. platensis*. Therefore, the appropriate addition method and concentration of exogenous selenium can promote the growth, organic materials accumulation, antioxidant activity, and photosynthetic performance of *A. platensis*.

*Arthrospira* has been called "the best health care product and the most ideal food for human beings in the 21st century" by the World Health Organization because of its extremely rich nutrients and physiological active substances. We aim to obtain more high-value-added products, achieve enrichment of mineral element nutrition, and improve the nutritional and health value of *A. platensis*. It is widely used in food additives and is expected to be applied in new types of health food in the future.

**Author Contributions:** Validation, S.Y.; Formal analysis, X.J., X.Z. and F.G.; Investigation, Z.W.; Data curation, Z.G., Y.Y. and Q.M.; Writing—original draft, W.W.; Writing—review & editing, D.G. All authors have read and agreed to the published version of the manuscript.

**Funding:** This study was supported by the National Natural Science Foundation of China (No.42367057, 31660151), the Ordos City science and technology plan project (No.2022YY023), and the Inner Mongolia Autonomous Region science and technology plan project (No.2022 YFXZ0037).

**Data Availability Statement:** Data are contained within the article.

**Conflicts of Interest:** The authors declare no conflict of interest.

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
