# Peer review of "Effects of Sodium Selenite on the Growth and Photosystem II Activity of Arthrospira platensis Gom."

_water, doi:10.3390/w16010020_

Round 1

Reviewer 1 Report (Previous Reviewer 1)

Comments and Suggestions for Authors

1. The object of research is a cyanobacterium, but the text contains the definition “algal” (lines 125, 126, 161, 163, 175).

2. Specify what type of artificial lighting and its parameters was used (line 129).

3. On line 133 the section 2.2.3.1 is mentioned, which is absent in the experimental part.

4. Fig 3.9 – replace the designation L (left) and R (right) with designation a and b

5. According to Fig. 3b, the maximum content of C-phycocyanin was about 45 mg/g or 4.5%. How can you explain such a low content of С-PC, since according to literature data its content reaches 15-20%?

Author Response

Dear reviewer,

I have revised the manuscript according to your suggestions. Thank you very much for your suggestions.

Have a good day.

Donghui

Reviewer 2 Report (Previous Reviewer 2)

Comments and Suggestions for Authors

All the comments I made earlier have been corrected.

Author Response

Dear reviewer,

I have revised the manuscript according to your suggestions. Thank you very much for your suggestions.

Have a good day.

Donghui

Reviewer 3 Report (New Reviewer)

Comments and Suggestions for Authors

We are dealing with an interesting and quite high-quality study of the physiological and biochemical reactions of the cyanobacteria A. platensis to the influence of sodium selenite. The work obtained results that definitely have promising practical applications.
The results and methods of work are presented quite clearly and comprehensively, with the necessary detail.
The reviewer has certain complaints about the “Discussion” and “Conclusion” sections.
Comments have been written in the form of notes to the text in corresponding places.
The article can be accepted for publication after correcting the noted points.

Author Response

Dear reviewer,

I have revised the manuscript according to your suggestions. Thank you very much for your suggestions.

Have a good day.

Donghui

Reviewer 4 Report (New Reviewer)

Comments and Suggestions for Authors

Consider that the species A. plantensis is commercially known as spirulina and the presence of synomens of the species, like Spirulina platensis and A. fusiformis, as well the taxonomic classification in Cyanophyta.

Page 2 line 98 change A. Platensis in A. platensis

Evidence in all the MN that the content of chlorophylls is only related to chlorophyll A

The sentence where ODck is the absorbance value of the light control, ODs is the absorbance value of the sample tube, VT is the total volume of the extraction solution (mL), VS is the volume of the extraction solution at the time of determination (mL), W is the biomass of the extracted enzyme solution of the A. platensis (g/L), and P is the protein content of the A. platensis cells (mg/L). is repeated in the same page 5 and partially again in lines 229-230.

Page 12 line 435 the term Purple Chlorella is probably referring to the common name of the organism and therefore it should not written in italic

Page 13 line 482 check the names of the species

It should be considered the current debate and alarm about the accumulation of substances in cyanophyta, as evidenced in the case of sulphites, and therefore the possibility that selenium could accumulated during the growth.

Phycocyanin is phycocyanin-C (C-PC)?

Is should be considered that as reported by Jaeschke et al. in Food Research International 2011, 143: 110314, the conditions of extraction are very important in the yield and stability of phycocyanins

Please check the presence of typing errors in Mn, like in page 3 line 135

In page 12 the sentence It was found from the research reports that different algae had different enrichment and tolerance of sodium selenite, but the overall performance was low promotion and high inhibition. should be supported by references and studies. Furthermore, the arguments is about algae or cyanobacteria?

Comments on the Quality of English Language

several sentences need revision and could simplified for a better understanding

Author Response

Dear reviewer,

I have revised the manuscript according to your suggestions. Thank you very much for your suggestions.

Have a good day.

Donghui

Round 2

Reviewer 3 Report (New Reviewer)

Comments and Suggestions for Authors

The authors changed some things in the article according to the comments. However, the main comments were ignored by the authors. The main comments were concerning the section “4.3. Effect of Na2SeO3 on the chlorophyll fluorescence of the A. platensis" and "Conclusion".
The remark regarding section 4.3 was that it is advisable to compare the acquired data with existing literature, including investigations conducted on other objects. Or have these data been obtained for the first time by the authors?
Regarding the conclusion section, the remark was that the conclusion appears incomplete. The work's primary findings should be presented in Conclusion, but it should also include an explanation of potential directions for future research and potential uses for the findings in practical applications. In this instance, it will be easier to see and more appealing to the value of the interesting outcomes of the job.
The authors made no changes to the article in response to these comments. The text of the article remains the same. This must be done. Only after this is it possible to consider publishing the manuscript.

Author Response

Dear reviewer,

I have revised the manuscript according to your suggestions. Thank you very much for your suggestions.

Have a good day.

Donghui

Reviewer 4 Report (New Reviewer)

Comments and Suggestions for Authors

There are several aspects needing careful consideration.

Check the meaning and the grammar of the sentence Some scholars have also reported that the structure of A. platensis changed under high concentration selenium stress, such as thylakoids was compacted, and cell wall was disintegration.

I suggest the author to evidence the current interest in spirulina, which is considered the best food of the future in consideration of its content in basic constituents and therefore improve the interest of the reader.

In 2.1. Experimental materials it is necessary an identification of the species, at least with a microscopic analysis, as required by the international scientific rules. It should also reported some information about how the material was obtained and treated.

It should be reported, also in respect of the reader and the cited literature that the species is well commercially known as spirulina, with a production and a consume of tons, In fact, the species was already known as S. platensis.

It should be clarified the meaning of phycocyanin and chlorophyll contents. For instance, probably chlorophylls is chlorophyll a.

Author Response

Dear reviewer,

I have revised the manuscript according to your suggestions. Thank you very much for your suggestions.

Have a good day.

Donghui

This manuscript is a resubmission of an earlier submission. The following is a list of the peer review reports and author responses from that submission.

Round 1

Reviewer 1 Report

Comments and Suggestions for Authors

1.     What is the novelty of this study? How the results obtained by the authors are consistent/different from previously obtained results? For example,

1)    Zhi-Yong Li, Si-Yuan Guo, Lin Li – Bioeffects of selenite on the growth of Spirulina platensis and its biotransformation. Bioresource Technology, 2003, 89 (2), 171-176.doi:10.1016/S0960-8524(03)00041-5.

2)    Tian-Feng Chen, Wen-Jie Zheng, Yum-Shing Wong, Fang Yang – Selenium-induced Changes in Activities of Antioxidant Enzymes and Content of Photosynthetic Pigments in Spirulina platensis. Journal of Integrative Plant Biology 2008, 50 (1): 40–48. doi: 10.1111/j.1744-7909.2007.00600.x

3)    ZHU Bao-Hua, SHEN Han, ZHU Zhao-Xia, et al. Influence of Selenium on Growth and Carbon Fixation Efficiency of Spirulina platensis. Periodical of Ocean University of China, 2019, 49 (11): 21-28. 10.16441/j.cnki.hdxb.20180223

4)    Yaghoobizadeh, F., Rajabi Memary, H., Roayaei Ardakani, M. – Bioaccumulation and the Effect of Selenate Concentration on Growth and Photosynthetic Pigment Content of Spirulina platensis. Journal of Phycological Research, 2021. 5 (1), pp. 624-641. doi: 10.48308/jpr.2021.221811.1005

5)    Cepoi, L.; Zinicovscaia, I.; Chiriac, T.; Rudi, L.; Yushin, N.; Grozdov, D.; Tasca, I.; Kravchenko, E.; Tarasov, K. – Modification of Some Structural and Functional Parameters of Living Culture of Arthrospira platensis as the Result of Selenium Nanoparticle Biosynthesis. Materials 202316, 852. doi:10.3390/ma16020852

2.     In the introduction, the authors are focus on plants PSII. However, the photosynthetic apparatus of Arthrospira plathensis has significant differences from plants (for example, the presence of phycobilisomes etc). It may be recommended to rewrite this section in relation to the object of study or phototrophic microorganisms (microalgae) in general.

3.     A number of methods are described chaotically.

4.     Line 41 – «Arthrospira platensis is an ancient prokaryotic multicellular filamentous algae». Arthrospira is a filamentous cyanobacterium. https://www.algaebase.org/browse/taxonomy/#8550

5.     Lines 96-97 – How it has been proven that microorganism isolated from alkaline lakes of the Ordos Plateau in Inner Mongolia is Arthrospira platensis?

6.     Line 101 – Why the cultivation time is limited by the logarithmic (exponential) growth phase? If the effect of Na2SeO3 concentration on A.platensis growth is considered, it would be interesting to establish the maximum value of biomass (g/L) achieved in the stationary growth phase.

7.     Table 1 – There are no explanations for the choice of Na2SeO3 tested concentrations in article.

8.     Lines 112-113 – «the solutions were incubated in 250 mL triangular flasks. The culture conditions were as follows: artificial light at room temperature, the bottles were shaken 6–8 times per day». May be the authors means Erlenmeyer or conical flasks?

9.     Lines 117-118 – The stock Na2SeO3 solution was added on 1, 5, and 7 days. Explain why the medium was fed by Na2SeO3 exactly on this cultivation times (1, 5, 7 day)?

10.  Lines 131-130 – 2.2.5. Determination of the soluble protein and phycocyanin contents of the Arthrospira.

It would be logical at first describe the isolation process and then the quantity determination. Why only water-soluble proteins were considered, not total protein (the content of which exceeds 60% of dry weight)? It is necessary to add references about the C-phycocyanin (C-PC) isolation method, as well as to the equation for calculating C-PC concentration. In general, it is necessary to destroy the cell wall before intracellular pigments (C-PC or chlorophyll a) isolation (freeze-thaw cycles, ultrasound, enzymatically, etc.). The most commonly used equation is the equation by Bennett and Bogorad (Bennett A., Bogorad L. Complementary chromatic adaptation in a filamentous blue-green alga. J. Cell. Biol. 1973. 58. 419–435).

11.  The methods section does not mention how the parameters (Fv/Fm, ABS/RC, DIo/RC, ETo/RC, PI ,ABS) discussed in the section 3.9. (Comparison of the parameters of the fluorescence induction kinetic curves of the Arthrospira at different concentrations of Na2SeO3) were calculated.

12.  The Discussion should be added with a comparison of your own results with the literature.

Author Response

Dear reviewer

Thank you very much for your scientific suggestions on the problems in the manuscript. I have revised your suggestions one by one in the manuscript. If there are any inappropriations, your suggestions are welcome.

Have a good day

Reviewer 2 Report

Comments and Suggestions for Authors

Comments on water-2611709 entitled “Effects of Sodium Selenite on the Growth and Photosystem â…¡ Activity of Arthrospira platensis

In the presented study effects of sodium selenite on the growth and photosystem â…¡ activity of Arthrospira platensis are analyzed. The results are interesting, well-presented and has been carried out using different (including statistical) methods. But I have some questions and remarks dealing with this manuscript.

1) Line 12. Arthrospira platensis is not a class, but a species; it belongs not to microalgae, but to cyanobacteria.

2) Lines 3, 41. “Arthrospira platensis” should be corrected to “Arthrospira platensis Gom.”.

3) In “2.1. Experimental materials" should be provided the coordinates of Arthrospira platensis sampling. It should also be clarified by what methods the taxonomic position of the species was identified. Did the authors use molecular genetic methods? Are they sure that the strain being studied is definitely Arthrospira platensis?

4)  Line 136. The brand, company and country of manufacture of the spectrophotometer should be provided.

Author Response

(The authors gave the same response as above.)

Reviewer 3 Report

Comments and Suggestions for Authors

This is a very interesting study and has certain application prospects. The authors measured many important indicators, but lacks in-depth analysis. The reviewer suggests that authors conduct in-depth analysis, such as cell morphology and related mechanisms.

Abstract: Line 30 & Materials and methods: Table 1, line 221 et al. use mol/L instead of mg/L.

Line 131: Lipid content and composition are also important indicators, why were they not measured?

Figure 1: The vertical coordinate unit should be consistent with the content, and it is recommended to use mol/L.

Figure 3.8: The resolution of the figure is low.

Discussion: Lack of in-depth analysis, which can also be combined with comparative analysis with other algae.

Comments on the Quality of English Language

no

Author Response

(The authors gave the same response as above.)

Round 2

Reviewer 1 Report

Comments and Suggestions for Authors

1) If you are talking about Arthrospira platensis Gom., then in the article itself, starting from 2.2. Experimental method use the generally accepted A.platensis. That is, replace Arthrospira with A.platensis throughout the text.

2) Unfortunately, many of the references are not publicly available

3) The clarifications and additions to a number of comments (1, 3, 4, 5, 7, 9, 10) does not added to article text

Comments 1:    

The comparison of the effect only on growth has been added and it is better to move this paragraph to the section Discussion

Comments 3:   

No changes made

Comments 4:     

Lines 42, 64, 66, 108, 175, 176, 185, 261, 427, 467, 484, 485.

 «Arthrospira platensis Gom. is an ancient prokaryotic filamentous algae» etc.

Comments 5.      

The corresponding reference (Qiao Chen, LI Bo-sheng, Zeng Zhao-qi. Alkaline Lakes and Spirulina (Arthrospira) resources in sandy land of Erdos, Journal of Arid Land Resources and Environment, 2001, 15(4): 86-91) has not been added to article text. Unfortunately, the article indicated in the response to the comments is not available for free access and it is impossible for readers of this article to understand how the testing microorganism was identificated, and what one of the four recognized types of Spirulina or Arthrospira was investigated in this work.

I would like to draw close attention to the fact that “Based on phenotypic data, as differences in helicity and trichome size, cell wall structure and pore pattern, gas vesicles, thylakoid pattern, trichome motility and fragmentation, combined to molecular information; is now accepted that Arthrospira Stizenberger ex Gomont, 1892 and Spirulina Turpin ex Gomont, 1892 are two distinct genera” https://doi.org/10.1016/j.algal.2023.103164

Comments 7.    

“The Na2SeO3 treatment concentration was determined by pre-experiments and literature reports.” But there are no added references (literature reports) – line 135.

Comments 9. 

The explanation was not added to text. It is advisable to indicate with arrows in the figure 3.1 the time of sodium selenite supplementation.

Comments 10. 

Reference on the national standard method for PC content determination was not added.

Question: Why only water-soluble proteins were considered, not total protein (the content of which exceeds 60% of dry weight)?

Answer: Soluble nitrogen and total nitrogen have a certain correlation, and each parallel sample only 200ml algal liquid, algal amount is difficult to meet the determination of total nitrogen, so the determination of soluble nitrogen.

What does dissolved and total nitrogen have to do with it? According to the described methods, the protein content was determined according to Bradford.

Comments 12.

Response 12: I added some relevant content in the discussion.

Lines 440-450. Reference 32 - There is no results for search by article title; in FOOD SCIENCE AND TECHNOLOGY for 2022 there is no such article at all https://ifst.onlinelibrary.wiley.com/loi/26891816/year/2022

4) Lines 468-472 “This indicated that too-high concentrations of Na2SeO3 stress led to reduced defense abilities of the antioxidant enzyme systems in the Arthrospira cells, as well as the excessive accumulation of H2O2 and the oxidation of the cell membranes by the ROS and the formation of lipid peroxidation products, which damaged the Arthrospira cells [39]”.

39. Song,C.; Chen,Q.; Wu,X.; Zhang,J.; Huang,C. Heat stress induces apoptotic-like cell death in two Pleurotus species. Curr Microbiol. 2014, 69, 611-6.

You need to justify the comparison of your own results with the finding that apoptosis-like cell death occurs in mycelium of Pleurotus in response to the heat stress.

It should be recalled that Pleurotus species (Basidiomycota, Agaricales) are worldwide-distributed white-rot fungi often responsible for causing hardwood decay in terrestrial ecosystems

5) Lines 174, 178, 184, 187. It is necessary to indicate the brand of the centrifuge and the type of rotor, or only the brand of the centrifuge and convert rpm in g.

6) Lines 19, 419-420 – “and we explored the optimal culture conditions”. Where is this mentioned in the article?

7) In the introduction there is a large section about selenium-enriched A.platensis. As part of the this research, the process of selenium accumulation in A.platensis cells and selenium content in biomass was not studied.

Reviewer 3 Report

Comments and Suggestions for Authors

Thank you for the revisions made by the authors. I have no further comments.